# Reasons for Pack Size Purchase among US Adults Who Purchase Cigars

**DOI:** 10.3390/ijerph18157790

**Published:** 2021-07-22

**Authors:** Jessica L. King, Anna Bilic, Julie W. Merten

**Affiliations:** 1Department of Health & Kinesiology, University of Utah, 250 S 1850 E, Salt Lake City, UT 84112, USA; u1309558@utah.edu; 2Department of Public Health, Brooks College of Health, University of North Florida, 1 S UNF Dr Building 39, Jacksonville, FL 32224, USA; jmerten@unf.edu

**Keywords:** tobacco regulatory science, cigars, tobacco control, tobacco packaging

## Abstract

With municipalities across the US establishing minimum cigar pack size regulations, it is critical to understand what drives pack size preference. The purpose of this exploratory study was to identify reasons for cigar pack size purchase. We used Amazon’s Mechanical Turk to survey adults who had purchased cigars and reported past 30 day use. Participants responded to an open-ended item asking their reasons for purchasing their usual pack size. Responses were double-coded and categorized. Of 152 respondents, 61 used traditional cigars, 85 used cigarillos, and 36 used filtered cigars. Across all cigar types, most participants (73.7%) purchased boxes rather than singles; 5–9-packs were the most popular pack size category (19.7%), followed by 20+-packs (18.4%). We identified 16 reasons for pack size purchase across seven categories: price, consumption, social aspect, convenience, product characteristics, availability, and general preferences. Reasons varied according to whether the consumer purchased larger or smaller pack sizes. In this exploratory study to identify reasons for cigar pack size purchases, findings were consistent with those identified through tobacco industry documents and in the cigarette literature. Future research should examine the prevalence of these reasons, including as a function of demographic and use characteristics, to help inform the understanding of potential minimum cigar pack regulations.

## 1. Introduction

Cigar use remains a public health concern, with nearly 12 million adults in the United States (US) reporting past 30 day use in 2019 [1]. In the US, cigar packages come in at least 12 different sizes ranging from singles to 200-packs [2]. It is unclear what drives cigar pack selection, although limited studies from the cigarette literature and tobacco industry documents provide some insight. It is well established that cigarette pack size is associated with the number of cigarettes smoked per day [3,4,5,6]; those who smoke more tend to purchase larger packs [3,4,5,6,7,8,9]. It is unclear whether this reflects preference or availability, as reported consumption often aligns with standard pack size (e.g., smoke “a pack” or 20 cigarettes per day) [3,5,6]. When a wider range of pack sizes are available, some suggest purchase behavior is used to regulate consumption, such that individuals will pay a premium for smaller packs [10]. Pack size is also tied to price [8,9]. Smaller packs are overall less expensive and have a lower barrier to entry [11]; larger packs have a lower price per stick and are perceived as more economical per unit [11,12]. Industry documents highlight other reasons for cigarette pack size preferences; smaller packs are perceived as easier to carry and conceal, whereas larger packs hinder the ability for individuals to reduce their smoking and have been used by the industry to retain customers [12].

Over 200 municipalities within the US regulate cigar pack size, and the US Food and Drug Administration (FDA) has authority to enact a product standard to establish a minimum pack size for cigars. With the potential for additional local, state, and federal regulations, it is critical to identify reasons consumers purchase a particular pack size to understand how policies may influence use and help identify unintended consequences. Given the paucity of cigar-specific research in this area, this exploratory study aimed to identify reasons for pack size purchase. Cigar-specific research is needed to understand whether the factors identified for cigarettes apply to cigars, identify unique factors for cigars, and provide cigar-specific evidence to inform FDA, local, and state regulation.

## 2. Materials and Methods

In April 2021, we recruited adults who reported ever purchasing large cigars, cigarillos, or little cigars and past 30 day cigar use from Amazon’s Mechanical Turk (MTurk). MTurk is an online marketplace commonly used for data collection in social science and tobacco control research [13,14,15]. As part of a larger study examining a variety of tobacco-related behaviors, we invited MTurk Workers residing in the US who were over the age of 21 to complete a survey. We restricted participation to MTurk Workers with high approval ratings (i.e., >90%) with at least 50 completed tasks, and participants received 1.25 USD.

Because patterns of use differ by cigar type, we categorized participants into traditional cigars, cigarillos, and filtered cigars by asking participants to select the cigar products they usually smoke from a photo of cigar types [16]. Those who selected multiple categories were asked whether they purchased different pack sizes on the basis of the cigar type. Regardless of the response to the latter question, participants who reported multiple cigar types were asked questions separately for each cigar type. Two questions assessed pack size purchasing behavior: (1) “Do you usually buy [product] by the box or pack, or as single cigars?” (2) “How many [product] come in the box or pack that you usually buy?” [16]. Participants were asked an open-ended item for pack size reasons: “What are the primary reasons you purchase the [product] pack size you typically purchase?”

Descriptive analyses characterized the sample. After initial review of all responses to the item on reasons for purchase, 26 codes were identified. Two trained coders independently coded all responses (interrater reliability = 86.9%). A third team member reviewed the results and determined the final code consulting with the full team in areas of discrepancy.

## 3. Results

### 3.1. Sample Characteristics

A total of 228 participants were eligible, and 216 completed cigar-related items. After reviewing open-ended responses for potential bots and inappropriate responses, 64 were deleted, resulting in 152 (66.7%) for analysis. Mean age was 33.9 (range 22–66), 71.1% identified as male, 75.7% identified as White, 16.4% identified as Black, 73.0% had a bachelor’s degree or higher, and 89.5% reported using another tobacco product within the past 30 days (Table 1). Twenty-six participants used multiple types of cigars, and all reported purchasing different pack sizes depending on the type of cigar used. Of those reporting using traditional cigars (*n* = 61), 70.5% purchased by the box, with 5–9-packs being the most prevalent size at 23.0%. For cigarillos (*n* = 85), 72.9% purchased by the box, with 5–9-packs and 20+-packs being the most prevalent at 18.8% each. For filtered cigars (*n* = 36), 75.0% purchased by the box, with 20+-packs being the most prevalent (27.8%).

The 182 open-ended responses (accounting for the 26 individuals using multiple cigar types) were coded into the following categories, presented on the basis of whether the response pertained to larger or smaller pack size preferences as defined by the respondent: (1) price, (2) consumption, (3) social aspect, (4) convenience, and (5) product characteristics. Remaining reasons for a particular pack size included general preference (e.g., “It is what I am used to”) or availability. Overviews of each are presented below; representative quotes are in Table 2.

### 3.2. Reasons for Purchasing Larger Pack Sizes

Comments on the savings per cigar were common among those who purchased larger packs (typically quantities from 5–50). Pack size was also attributed to consumption, with respondents purchasing larger packs because they typically consumed that amount within a week. The social aspect refers to respondents across cigar types purchasing larger packs to share with friends and family. Convenience was commonly mentioned as a reason for purchasing larger packs. Several participants noted that they purchased more to reduce the number of trips to the store. Others noted that they purchased extra to have around just in case they wanted to smoke or if the product was not available in the future. Few respondents stated product characteristics as reasons for purchasing larger packs. Some filtered cigar purchasers mentioned that their packages were easy to carry, while one cigarillo purchaser noted that “the box will save the cigars from the water” suggesting that the larger pack came with a more durable package.

### 3.3. Reasons for Purchasing Smaller Pack Sizes

Price was a frequently stated reason for purchasing smaller packs, with comments focused on the overall lower price and ability to save money, primarily among those who typically purchased singles. Regarding consumption, several participants reported purchasing smaller packs because they did not smoke often or to regulate or reduce their smoking. For example, a participant who typically purchased single cigarillos said “because it is not often I smoke one, …, so I do not want to buy more than that as I am trying to cut down.” The social aspect was introduced among several participants who purchased singles or two-packs: “I am usually buying one for me and one for a close friend when we get together.” Smoking alone was also a reason for purchasing smaller packs: “Most times I smoke alone so I do not see reasons buying in large packs.” Convenience was not a common reason for purchasing small packs. One participant stated that “it is a lot more convenient to just have a small number around” which may refer to storage, discussed in more detail below.

Several product characteristics were identified as reasons for purchasing smaller packs, including freshness, switching, and storing product. Participants noted that they preferred smaller packs because it allowed for switching to other tobacco products or new flavors/brands of cigars. Others preferred smaller packs to ensure freshness. Storing cigars was a common reason for purchasing smaller cigars, as participants noted they did not have a humidor, and one mentioned purchasing singles because, with a child around, they did not like storing cigars.

## 4. Discussion

In this exploratory study to identify reasons for cigar pack size purchases, we surveyed a convenience sample of 152 adults who consumed cigars in the previous 30 days. Respondents identified 16 specific reasons for purchasing particular cigar pack sizes, consistent with reasons that have been cited in industry documents or cigarette experiments. We also identified novel factors including the social aspect, child safety concerns, and storing the product. 

Our findings identify the importance of price, consumption, convenience, and product characteristics as reasons for purchasing a particular pack size. Increased price is an intended consequence of minimum pack size regulations. The overall cheap price was a commonly reported reason for purchasing singles while savings per stick was a primary reason for purchasing larger packs. Participants reported purchasing smaller packs to regulate or reduce their smoking. This finding has been acknowledged elsewhere [10,12,17,18]. In response, the industry has used larger packs to keep people from quitting and to maintain sales volumes even as the number of people who smoke continues to decrease [12]. With a minimum pack size regulation, this population might stop purchasing cigars (intended consequence), purchase a larger pack which may lead to consuming more (unintended consequence), seek out product in neighboring jurisdictions, switch to another product, or share with friends. 

The social aspect was identified among participants across pack size preferences. Responses implied that participants purchased additional product to share or to fit in. Peer influence is a critical factor for tobacco use [19], although its implications for purchasing are less known. Few mentioned small packs as convenient or easy to carry, which has been highlighted as appealing to young adults in industry studies [12]. However, in the current study, several participants noted that their 20-packs were easy to carry and the boxes were ideal to store the product, presumably compared to a foil package or single large cigar that might be sold without a box. Considering product storage, highlights include concerns about safety and having product near children, as well as purchasing small packs to avoid having stale product. The latter finding is similar to industry studies that found that large packs could potentially detract from perceived quality.

Participants noted that they preferred smaller packs because they wanted to try new flavors and brands of cigars or alternate between tobacco products. Small packs have been used by the tobacco industry to entice users of competitor brands or to provide novel product to their current brand users. The industry has also used cross-product promotions to extend brand loyalty [20]. While we did not explore findings by demographic characteristics, a recent analysis of demographic and use characteristics by pack size preference found that those who purchase smaller packs tended to be younger and more likely to use other tobacco products [21]. This population, likely experimenting with products, will be critical for monitoring the impact of minimum pack regulations to ensure unintended consequences are minimal. 

These findings are subject to several limitations. Data were from a convenience sample that was highly educated, predominantly White, and male; moreover, compared to analyses from the Population Assessment of Tobacco and Health nationally representative study, our sample had a higher prevalence of purchasing cigars by the box [8]. Thus, the frequencies for which we identified reasons for use should not be interpreted as representative. We identified inconsistencies in open-ended and multiple-choice responses on some items, which increased concerns about data quality and led to a smaller sample than originally intended with the inability to explore reasons for purchase based on demographic or use characteristics. Finally, we were unable to probe or follow up to obtain additional clarification on responses. While our interpretations strived for objectivity, it is possible we misinterpreted the reasons some participants purchase particular pack sizes. 

## 5. Conclusions

This study identified reasons consumers purchase a particular cigar pack size and represents an initial step in informing public health decision-making around cigar pack policies. We identified 16 reasons for purchasing cigars based on pack size, which extends understanding of how the industry has used pack size and findings from the cigarette literature to cigars. Future studies with nationally representative samples are needed to inform the prevalence of these reasons and examining differences in reasons for purchase by demographic and use characteristics might provide greater insight into whether potential minimum pack size policies would have an equitable impact.

## Figures and Tables

**Table 1 ijerph-18-07790-t001:** Sample characteristics.

	Full Sample	Traditional Cigars	Cigarillos	Filtered Cigars
	*N* = 152	*N* = 61	*N* = 85	*N* = 36
**Age (mean (SD))**	33.9 (8.4)	35.9 (9.5)	33.5 (9.1)	33.9 (8.6)
**Sex**				
Male	71.1% (108)	68.9% (42)	71.1% (59)	66.7% (24)
Female	25.0% (38)	24.6% (15)	27.7% (23)	33.3% (12)
Other	1.4% (2)	1.6% (1)	1.2% (1)	0
**Race and Ethnicity ^1^**				
White only	67.8% (103)	60.7% (37)	70.6% (60)	66.7% (24)
Black only	15.1% (23)	18.0% (11)	16.5% (14)	19.4% (7)
Asian only	2.0% (3)	1.6% (1)	2.4% (2)	2.8% (1)
Multiple	5.3% (8)	8.2% (5)	3.5% (3)	0
Hispanic	5.9% (9)	3.3% (2)	4.7% (4)	11.1% (4)
**Education**				
HS degree	8.6% (13)	3.3% (2)	11.8% (10)	8.3% (2)
Some college	15.1% (23)	9.8% (6)	18.8% (16)	5.6% (2)
BS degree or higher	73.0% (111)	80.3% (49)	67.0% (57)	86.1% (31)
**Usual purchase size ^2^**				
Single	26.3% (40)	29.5% (18)	27.1% (23)	25.0% (9)
2–4-pack	11.2% (17)	9.8% (6)	17.6% (15)	11.1% (4)
5–9-pack	19.7% (30)	23.0% (14)	18.8% (16)	19.4% (7)
10–19-pack	15.8% (24)	19.7% (12)	17.6% (15)	16.7% (6)
20+-pack	18.4% (28)	18.0% (11)	18.8% (16)	27.8% (10)

Notes: ^1^ The race category also included American Indian and Alaska Native, Native Hawaiian and Pacific Islander, and Other, but no respondents reported these categories. ^2^ Thirteen respondents had different pack size groups across cigar types and are not included in the total response column.

**Table 2 ijerph-18-07790-t002:** Reasons for pack size purchases.

Category	Subcategory	Representative Quote (Cigar Type, Pack Size)
Price	Cheaper per stick	Save money by buying a box vs buying single cigars (TC ^1^, 20)
	Cheaper overall	They are cheap (FC ^2^, 1)
Consumption	Regulate consumption	Buying only 1 helps me from smoking too much (FC, 1)
	Matches consumption	Lasts me about a week (CG ^3^, 10)
	Do not consume often	Because I only use maybe one a day at most (CG, 1)
Social	Share with friends	I usually share them with some friends and family so a pack makes more sense (CG, 4)
	Friend preference	My friends prefer this (FC, 6)
	To smoke alone	Most times I smoke alone so I do not see reasons buying in large packs (TC, 1)
Convenience	Fewer store visits	So that I do not have to make a trip for many days to restock; better value as well for the money (CG, 5)
	Have when needed	I like to have more around; better value too (TC, 10)
Product Characteristics	Like to switch products	I like to switch flavors instead of committing to one flavor (TC, 1)
	To ensure freshness	I prefer them fresh so I would rather buy smaller boxes so I know I finish them before they go stale (CG, 4)
	Package for storage	Because they are in a package to hold them and not get bent before use; also, so that I do not have to go back to the store later I buy packs (CG, 5)
	Child safety	I do not like storing cigars because I have a child always around me (CG, 1)
	Easy to carry	It is compact to carry in pockets (FC, 8)
Availability	It’s available	It only comes in a 5-pack box; they stopped selling singles (FC, 5)
General preference	No specific reason	I always prefer to purchase singles rather than purchasing a package (CG, 1)

^1^ TC = traditional cigars (*N* = 61); ^2^ FC = filtered cigars (*N* = 36); ^3^ CG = cigarillos (*N* = 85).

## Data Availability

Data are available by contacting the corresponding author.

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
