# Peer review of "Reasons for Pack Size Purchase among US Adults Who Purchase Cigars"

_ijerph, 2021, doi:10.3390/ijerph18157790_

Round 1
Reviewer 1 Report
The authors present a descriptive study of cigar purchasing behaviors in a convenience sample of cigar users in the United States. Participants were selected from Amazon Mechanical Turk workers. Self-reported reasons for the size of cigar "packs" purchased were reported. Most participants reported purchasing cigars in boxes rather than singles, and in smaller packs of 5-9. Reasons for purchasing included price, consumption, convenience, product characteristics, and availability. The authors conclude that the reasons for pack size purchase of cigars are similar to reasons for purchasing cigarette pack sizes. Overall, this is an interesting, well-written study that presents important insights into the reasons for purchasing cigar packs of different sizes. I have some minor questions/clarifications for the authors.
Major criticisms:
- In the introduction, to help frame the problem, it could be useful to insert something about the known prevalence of cigar smoking in the US. If possible, the authors could identify the prevalence of exclusive cigar use and dual use with cigarettes. This information will provide additional context for the study. While exclusive cigar use might be low, this additional information provides context for proposed regulations based on pack size (e.g., importance of minimum pack sizes for filtered cigars and cigarillos.)
- Page 1, line 44: "Over 200 municipalities within the US regulate pack size" -- specify whether you mean cigarette pack size or cigar pack size. I'm guessing it's cigar pack size, given the previous paragraph, but a simple qualifier helps distinguish when you mean cigarette pack size and when you are referring to cigar pack size.
- Page 2, line 53: sample of MTurk workers -- please clarify whether the sample includes past-30 day cigar users who purchased any cigars in the past 30 days? Also clarify whether cigars were broadly defined as traditional cigars, filtered cigars, and cigarillos. It would also be useful to clarify whether participants were exclusive cigar smokers who do not smoke cigarettes.
- Line 56: "MTurkers" -- this terms seems somewhat colloquial. Perhaps use "MTurk workers" instead?
- Page 2, line 78: "After reviewing open-ended responses for potential bots" -- this seems to contradict the information presented in lines 57-59, where the sample was restricted to workers having high approval ratings with at least 50 completed tasks. Was there no way to filter possible bots at the outset?
- In Table 1, it might be useful to separate those participants purchasing multiple products, so that categories are mutually exclusive. For those purchasing multiple products, report combinations in the text, if there were not too many combinations or perhaps report the dominant type.
- Table 1, usual purchase size row: Maybe split this into a separate table, by whether purchased single or multiple products. This provides a better accounting of the data.
- Page 3, line 89: For the 181 open-ended responses -- is it the case that open-ended responses were provided for all cigar types? If so, report how many of the n=152 participants reported a single type and how many reported multiple types, i.e., how do you get from the 152 unique workers retained to 181 open-ended responses?
- Table 2, footnote: n=36 for filtered cigars while table 1 lists n=35. Should the footnote read n=35?
- Page 5, line 137: It would be better to state that reasons for cigar pack size purchasing are consistent with cigarette pack size purchases, not confirmed. This is only a single exploratory study, and a small one at that (which is OK, it is what it is, and an initial look is important. But more data, esp. representative data such as PATH, would go a long way to demonstrate that reasons for pack size purchasing of cigars are similar to cigarettes). Same for line 140 -- would not state "confirm" here.
- Line 152: "The social aspect was introduced among participants" -- suggest changing "introduced" to "identified"
- Lines 159-161 are somewhat redundant, as this information has already been mentioned. However, I appreciate the difficulty, since this deals with the more qualitative, open-ended text. Perhaps try to strike a balance in the information presented without repeating too much of the information?
Minor comments:
- Page 1, line 43: rephrase "larger packs counter objectives to reduce smoking" -- I think I know what you mean, but perhaps rephrase to increase clarity.
- Page 2, line 65: suggest adding "purchasing" as a qualifier, i.e., "Two questions assessed pack size purchasing behavior"
Author Response
Response: Thank you for the constructive feedback. We truly appreciate it!
The authors present a descriptive study of cigar purchasing behaviors in a convenience sample of cigar users in the United States. Participants were selected from Amazon Mechanical Turk workers. Self-reported reasons for the size of cigar "packs" purchased were reported. Most participants reported purchasing cigars in boxes rather than singles, and in smaller packs of 5-9. Reasons for purchasing included price, consumption, convenience, product characteristics, and availability. The authors conclude that the reasons for pack size purchase of cigars are similar to reasons for purchasing cigarette pack sizes. Overall, this is an interesting, well-written study that presents important insights into the reasons for purchasing cigar packs of different sizes. I have some minor questions/clarifications for the authors.
Major criticisms:
- In the introduction, to help frame the problem, it could be useful to insert something about the known prevalence of cigar smoking in the US. If possible, the authors could identify the prevalence of exclusive cigar use and dual use with cigarettes. This information will provide additional context for the study. While exclusive cigar use might be low, this additional information provides context for proposed regulations based on pack size (e.g., importance of minimum pack sizes for filtered cigars and cigarillos.)
Response: We added a statement on the prevalence of cigar use in the US. Unfortunately, national data on patterns of cigar use among those who smoke cigars is dated (2013-14 vs 2019 for overall prevalence), so we aren’t able to report more detailed information.
- Page 1, line 44: "Over 200 municipalities within the US regulate pack size" -- specify whether you mean cigarette pack size or cigar pack size. I'm guessing it's cigar pack size, given the previous paragraph, but a simple qualifier helps distinguish when you mean cigarette pack size and when you are referring to cigar pack size.
Response: We clarified that we mean cigar policies.
- Page 2, line 53: sample of MTurk workers -- please clarify whether the sample includes past-30 day cigar users who purchased any cigars in the past 30 days? Also clarify whether cigars were broadly defined as traditional cigars, filtered cigars, and cigarillos. It would also be useful to clarify whether participants were exclusive cigar smokers who do not smoke cigarettes.
Response: We edited the sentence to now read we recruited adults who reported ever purchasing large cigars, cigarillos, or little cigars and past 30-day cigar use from Amazon’s Mechanical Turk. We also added a sentence on the rates of past 30 day otp use in results.
- Line 56: "MTurkers" -- this terms seems somewhat colloquial. Perhaps use "MTurk workers" instead?
Response: We edited this throughout.
- Page 2, line 78: "After reviewing open-ended responses for potential bots" -- this seems to contradict the information presented in lines 57-59, where the sample was restricted to workers having high approval ratings with at least 50 completed tasks. Was there no way to filter possible bots at the outset?
Response: Unfortunately, bots are an increasingly common problem on MTurk (e.g., 30% of responses in a batch are clearly bots), which is why we opted to review each response prior to analyzing the data. Many people using MTurk will automatically approve and pay rather than reviewing each response and rejecting poor quality responses. This allows some to have a high approval rating despite low quality responses. Additionally, Qualtrics provides a captcha score, but that’s not blocking them from taking the survey. Fortunately, we had several specific, open-ended items which allowed us to assess data quality a bit better than if we’d gone entirely with multiple-choice.
- In Table 1, it might be useful to separate those participants purchasing multiple products, so that categories are mutually exclusive. For those purchasing multiple products, report combinations in the text, if there were not too many combinations or perhaps report the dominant type.
Response: We considered this breakdown when initially reporting the results, and again discussed it after reviewing the comment. There are multiple combinations, with some respondents responding to all three product types, and it becomes quite wieldy to report. Ultimately, this is a convenience sample, it’s not intended to be representative or inform conclusions about the prevalence of use patterns, nor do we get into discussions across cigar groups. We believe this table is sufficient to inform the readers about the sample without leading to additional unintended conclusions.
- Table 1, usual purchase size row: Maybe split this into a separate table, by whether purchased single or multiple products. This provides a better accounting of the data.
Response: We don’t follow what the reviewer is asking for here, particularly with regard to better accounting of the data. We have presented the information on pack sizes, across cigar type to help provide some context to the findings. However, as noted in the previous response, this is a convenience sample that, as the limitations note, differs quite a bit from nationally representative samples, so, items on the prevalence are not intended to be interpreted or to inform conclusions.
- Page 3, line 89: For the 181 open-ended responses -- is it the case that open-ended responses were provided for all cigar types? If so, report how many of the n=152 participants reported a single type and how many reported multiple types, i.e., how do you get from the 152 unique workers retained to 181 open-ended responses?
Response: The methods state Those who selected multiple categories were asked whether they purchased different pack sizes based on the cigar type. Regardless of response to the latter question, participants who reported multiple cigar types were asked questions separately for each type.
The results further state Thirty-two (or 31??) participants used multiple types of cigars
We added additional clarification The 181 open-ended responses (accounting for the 32 individuals using multiple cigar types) were coded into the following categories.
- Table 2, footnote: n=36 for filtered cigars while table 1 lists n=35. Should the footnote read n=35?
Response: It is 36. We edited this throughout.
- Page 5, line 137: It would be better to state that reasons for cigar pack size purchasing are consistent with cigarette pack size purchases, not confirmed. This is only a single exploratory study, and a small one at that (which is OK, it is what it is, and an initial look is important. But more data, esp. representative data such as PATH, would go a long way to demonstrate that reasons for pack size purchasing of cigars are similar to cigarettes). Same for line 140 -- would not state "confirm" here.
Response: This is a really great point. Thank you. Changes made throughout.
- Line 152: "The social aspect was introduced among participants" -- suggest changing "introduced" to "identified"
Response: Edited as suggested.
- Lines 159-161 are somewhat redundant, as this information has already been mentioned. However, I appreciate the difficulty, since this deals with the more qualitative, open-ended text. Perhaps try to strike a balance in the information presented without repeating too much of the information?
Response: We rephrased this section to read as interpretation rather than repeating the results.
Minor comments:
- Page 1, line 43: rephrase "larger packs counter objectives to reduce smoking" -- I think I know what you mean, but perhaps rephrase to increase clarity.
Response: We edited the sentence to read larger packs hinder the ability for individuals to reduce their smoking and have been used as an industry strategy to retain customers.
- Page 2, line 65: suggest adding "purchasing" as a qualifier, i.e., "Two questions assessed pack size purchasing behavior"
Response: Edited as suggested.
Reviewer 2 Report
The manuscript presents an interesting issue related to reasons for cigars pack size purchase. This manuscript is well written and limitations of this research are precisely mentioned, however some aspects might be improved.
The introduction section should include information about the existence of similar studies. If this is the first publication of such research, it also should me mentioned. The precise aim of this study should also be emphasized at the end of introduction section.
In Materials and Methods section (line 64-65) there is information that "participants who reported multiple cigar types were asked questions separately for each type" and in results section that "32 participants used multiple types of cigars (line 81-82). I understand that this number of 152 is the number of analyses surveys, therefore the information of number of participants is missing and also should be included.
The results section 3.3 should be titled "Reasons for Purchasing Smaller Pack Sizes" instead of "Larger" (titles of sections 3.2 and 3.3 are the same).
I would also appreciate some graphic presentation of the results.
The conclusions section is concentrated on significance od this research in potential minimum pack size policies, however I believe that information collected in this study might also become useful in smoking cessation programs.
Author Response
The manuscript presents an interesting issue related to reasons for cigars pack size purchase. This manuscript is well written and limitations of this research are precisely mentioned, however some aspects might be improved.
Response: Thank you for the constructive feedback!
The introduction section should include information about the existence of similar studies. If this is the first publication of such research, it also should me mentioned. The precise aim of this study should also be emphasized at the end of introduction section.
Response: The introduction includes the following text: It is unclear what drives cigar pack selection, though limited studies from the cigarette literature and tobacco industry documents provide some insight. And in the second introductory paragraph: With the potential for additional local, state, and federal regulations, it is critical to identify reasons consumers purchase a particular pack size to understand how policies may influence use and help identify unintended consequences. Given the paucity of cigar-specific research in this area, this exploratory study aimed to identify reasons for pack size purchase. Cigar-specific research is needed to understand whether the factors identified for cigarettes apply to cigars, identify unique factors for cigars, and provide cigar-specific evidence to inform FDA, local, and state regulation.
In Materials and Methods section (line 64-65) there is information that "participants who reported multiple cigar types were asked questions separately for each type" and in results section that "32 participants used multiple types of cigars (line 81-82). I understand that this number of 152 is the number of analyses surveys, therefore the information of number of participants is missing and also should be included.
Response: There were no missing responses; there were 26 participants who reported multiple product use, and 182 total responses.
The results section 3.3 should be titled "Reasons for Purchasing Smaller Pack Sizes" instead of "Larger" (titles of sections 3.2 and 3.3 are the same).
Response: Edited as suggested.
I would also appreciate some graphic presentation of the results.
Response: We assume this refers to perhaps frequencies or a pie chart of the qualitative categories? We considered providing this, but given the limitations of the sample (e.g., convenience sample, bots), we chose to frame this as an exploratory study highlighting the reasons for use, rather than a focus on the prevalence of each of the reasons for use. Regardless of the caveats, if we shared prevalence for the statements, it is likely to be interpreted as prevalence for the reasons, which, we prefer to avoid.
The conclusions section is concentrated on significance od this research in potential minimum pack size policies, however I believe that information collected in this study might also become useful in smoking cessation programs.
Response: This is an interesting point and certainly something that might be worth discussing in perhaps a different setting. We aren’t aware of any cigar studies on reducing cigars for cessation. However, the existing literature on cutting back cigarettes indicates that it can hinder successful cessation. As such, we prefer to not speculate on what the implications would be for reducing pack size to help with cigar cessation until research on the benefits and harms is available.
Round 2
Reviewer 1 Report
The authors have answered and addressed all of my questions and criticisms.